# Multifaceted intervention for the prevention and management of musculoskeletal pain in nursing staff: Results of a cluster randomized controlled trial

**Mercè Soler-Font** [1,2], **José Maria Ramada** [1,2,3], **Sander K. R. van Zon** [4], **Josué Almansa** [4], **Ute Bültmann** [4], **Consol Serra** [1,2,3] *, **on behalf of the INTEVAL_Spain research team** [¶]

**1** Center for Research in Occupational Health, University Pompeu Fabra/ IMIM (Hospital del Mar Medical Research Institute), Barcelona, Spain, **2** CIBER of Epidemiology and Public Health, Madrid, Spain, **3** Occupational Health Service, Parc de Salut Mar, Barcelona, Spain, **4** Department of Health Sciences, Community and Occupational Medicine, University of Groningen, University Medical Center Groningen, Groningen, The Netherlands

¶ Membership of the INTEVAL Spain research team is provided in the Acknowledgments.
* consol.serra@upf.edu

**Data Availability Statement:** All relevant data are within the manuscript and its Supporting Information files.

## Abstract

### Background

Nurses and nursing aides are at high risk of developing musculoskeletal pain (MSP). This study aimed to evaluate a multifaceted intervention to prevent and manage MSP in two hospitals.

### Material and methods

We performed a two-armed cluster randomized controlled trial, with a late intervention control group. Clusters were independent hospital units with nursing staff as participants. The intervention comprised three evidence-based components: participatory ergonomics, health promotion activities and case management. Both the intervention and the control group received usual occupational health care. The intervention lasted one year. MSP and work functioning data was collected at baseline, six and 12-month follow-up. Odds ratios (OR) and their 95% confidence intervals (95%CI) were calculated for MSP risk in the intervention group compared to the control group using logistic regression through GEE. Differences in work functioning between the intervention and control group were analyzed using linear regression through GEE. The incidence of sickness absence was calculated through logistic regression and Cox proportional hazard modeling was used to analyze the effect of the intervention on sickness absence duration.

### Results

Eight clusters were randomized including 473 nurses and nursing aides. At 12 months, the intervention group showed a statistically significant decrease of the risk in neck, shoulders and upper back pain, compared to the control group (OR = 0.37; 95%CI = 0.14–0.96). A

**Funding:** AM and JM received funding by the Instituto de Salud Carlos III-FEDER, reference numbers PI14/01959 and PI17/00779, respectively (https://sede.isciii.gob.es/anouncements_detail.jsp?pub=10623). The funders had no role in the study design, data collection, analysis, decision to publish, or preparation of the manuscript.

**Competing interests:** The authors have declared that no competing interests exist.

**Abbreviations:** MSP, Musculoskeletal pain; MSDs, Musculoskeletal disorders; OHS, Occupational Health Service.

reduction of low back pain was also observed, though non statistically significant. We found no differences regarding work functioning and the incidence and duration of sickness absence.

## Conclusions

The intervention was effective to reduce neck, shoulder and upper back pain. Our results, though modest, suggests that interventions to prevent and manage MSP need a multifactorial approach including the three levels of prevention, and framed within the biopsychosocial model.

## Background

Musculoskeletal pain (MSP) is a common characteristic of musculoskeletal disorders (MSDs) [1]. MSP affects people across the life-course in all regions of the world, and it is known that between 20%–33% of people worldwide live with MSP [2]. There is evidence that around 70–80% of workers report discomfort due to awkward postures and forceful work [2, 3]. This is reflected in sickness absence. In Europe, MSDs represent 50% of sickness absences and 60% of permanent disabilities [4].

Healthcare workers, especially nursing staff, are an occupational group at high risk of developing MSP. The prevalence of MSP is higher among healthcare workers than among the mean of across Europe [4–6]. In Spain in 2016, 83% of healthcare workers reported MSP [3, 7]. Around 60–70% of healthcare workers are nursing staff i.e. nurses and nursing aides [3, 8]. They are an occupational group at a high risk for developing MSP and musculoskeletal injuries because of exposure to heavy manual lifting of patients. Nursing staff with daily patient-handling had almost twice the risk of developing work-related back injuries compared with nurses without daily patient-handling [8, 9]. MSP may also affect other occupational outcomes like work functioning. Work functioning is the worker's ability to meet the demands of work given a worker's health status [10, 11], and can be affected in workers with MSP [12]. Work functioning encompasses several dimensions of work, such as work scheduling demands, output demands, and physical, mental and social demands, all of which should be considered to manage MSP due its multi-causal etiology [13–16].

MSP is influenced by a complex and dynamic interaction between biological, psychosocial, cultural, individual and environmental factors [17, 18]. Thus, it requires the biopsychosocial approach that acknowledges that the level of pain and disability are a result of these complex interactions and that these relationships will determine how the person will be able to manage illness [19].

In recent years, several occupational interventions have been performed to reduce and prevent MSP and to promote early return to work after related sickness absence [20–26]. Multicomponent interventions that combine several specific approaches for various determinants have been shown to be more effective than those based just on one specific component [21–26]. Also, a recent systematic review recommended multifaceted interventions which include components as healthcare provision, coordination of services and work accommodation to improve MSP and reduce the time until return to work [21].

In this study, we chose the components covering all three levels of prevention. Primary prevention components focused on 1) occupational risk factors to protect healthy workers from MSP and sickness absence related to MSDs through participatory ergonomics and 2) the promotion of healthy lifestyles at work (e.g. physical activity, emotional well-being and healthy

diet). Secondary and tertiary prevention comprised a case management service to identify early MSP, improve prognosis and reduce the probability of sickness absence, and to allow a safe and sustainable return to work, respectively [27].

Therefore, INTEVAL_Spain was developed as an evidence-based intervention, which addresses all three levels of prevention. This study aimed to evaluate the effectiveness of the INTEVAL_Spain intervention. We hypothesized that a multifaceted intervention covering all three levels of prevention in nursing staff compared to usual care (1) reduces the prevalence of MSP, (2) prevents related sickness absence, and (3) enhances the work functioning.

## Methods

The CONSORT statement and the CONSORT extension for cluster randomized trials were used to describe the study design [28, 29]. More detailed information on the content of the intervention and the evaluation process has been described elsewhere [27].

### Study setting and participants

The study was designed as a two-armed cluster randomized controlled trial with a late intervention control group (i.e. the control group received the intervention after the study period). Two public healthcare institutions participated in the trial. Both were third-level hospitals located in Barcelona province (Spain), with specialized acute care, psychiatry, long-term care and primary care centers.

Clusters were independent hospital units, and participants were the nursing staff (nurses and nursing aides) working in these units. The clusters had a variable number of nurses and nursing aides (20 to 60) and were selected because of exposure to high physical demands i.e. patient-handling mobilizations, prolonged standing and adopting awkward postures; especially for nursing staff, according to the annual risk assessments of the Occupational Health Service (OHS) prior to the study.

All nursing staff from these units was eligible for the study, including those on sickness absence. Temporary nursing staff who had worked for short periods of time (less than 3 months), those who worked in several units and those who were on sabbatical leave were excluded from the study.

Informative briefings were held in each cluster before randomization, informed consents were collected and the baseline questionnaire was administered and completed before workers knew their unit assignment to the intervention or control group. The questionnaires were anonymous. Follow-up questionnaires were administered at six and 12 months (Fig 1).

### Intervention

The INTEVAL_Spain intervention aimed to prevent MSP and related sickness absence and comprised three components reflecting the three levels of prevention. These components were (i) participatory ergonomics as primary prevention of occupational risk factors; (ii) healthy lifestyle promotion program, also as primary prevention; and (iii) a tailored case management program as secondary and tertiary prevention. The implementation of components was progressive to facilitate participation and logistics, and overlapped (Fig 2). The entire intervention lasted one year, starting at the beginning of September 2016. During the first month, the participant's recruitment was carried out and the baseline questionnaire was completed. After cluster randomization, the intervention started a month later with participatory ergonomics as this intervention required a longer process to be fully implemented, e.g. purchase of materials. Case management was carried out from March 2017 to December 2017. The healthy lifestyle promotion program started in December 2016 with Nordic Walking, a healthy diet web

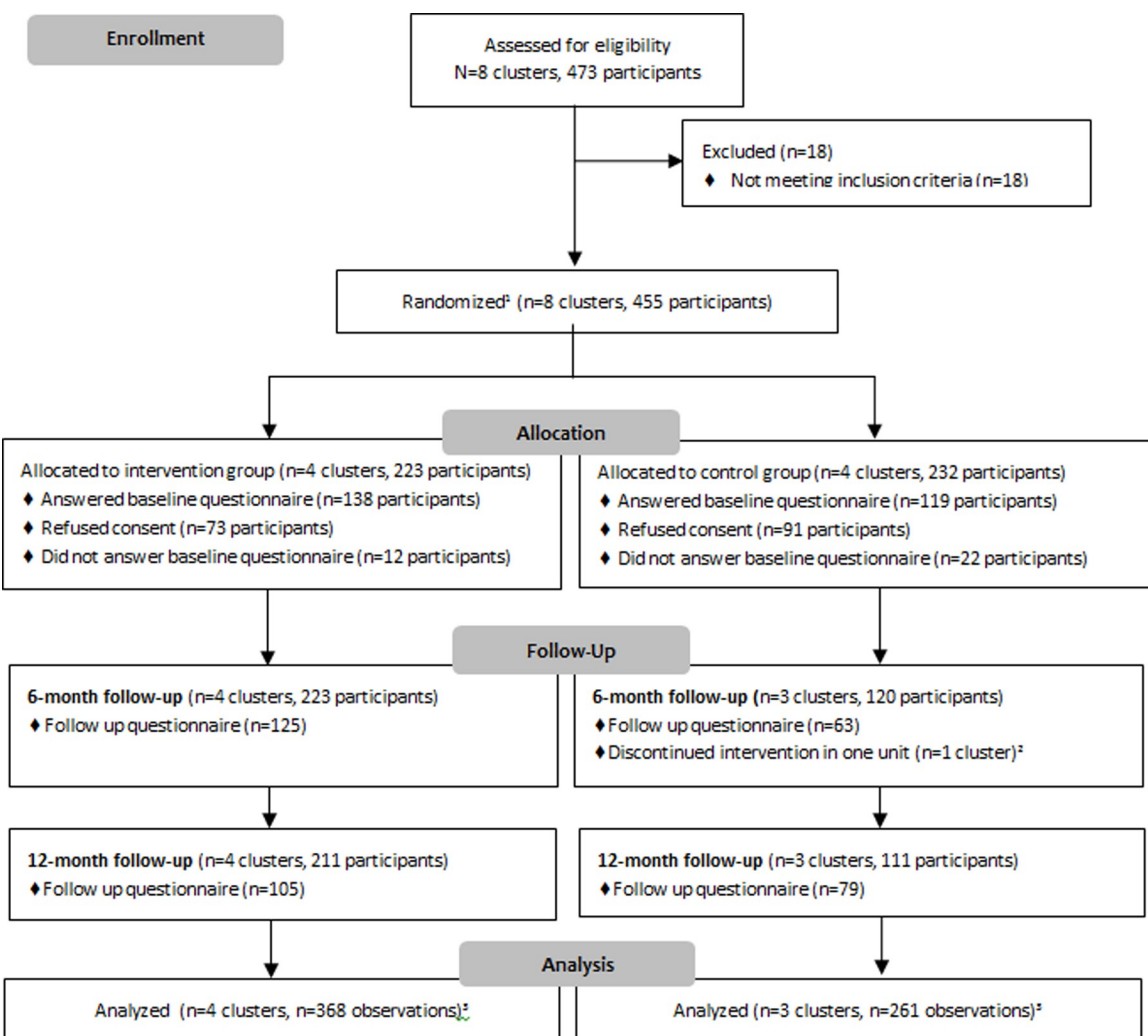

**Fig 1. Flow chart of participant recruitment, allocation and outcome assessment.** The Clinical Research Ethical Committee of Parc de Salut Mar provided the ethical approval of the study on July 10, 2014. The authors confirm that all ongoing and related trials for this intervention are registered. INTEVAL_Spain was retrospectively registered in ISRCTN (registration number 15780649). This trial was retrospectively registered because it was not a requirement for the approval of the protocol by the Ethical Committee of Parc de Salut Mar. A discrepancy between the text and the protocol concerns the secondary outcome Organizational Preventive Culture as this could not be analyzed because of missing data.

platform and mindfulness training. Follow-up questionnaires were administered at 6 and 12 months follow-up, during April and November 2017.

Participatory ergonomics was based on and adapted from the ERGOPAR Method [30], which was tested in Spanish companies with good results [31, 32]. Participatory ergonomics is a widely recommended approach to reduce MSP through reducing exposure to biomechanical and psychosocial risk factors, i.e. psychosocial stressors, social support, [33, 34] even though other researchers did not find consistent positive results [35, 36] probably due to different company dynamics. Participatory ergonomics was divided into three phases: diagnostic, treatment and implementation. It began with the diagnostic phase consisting of the administration

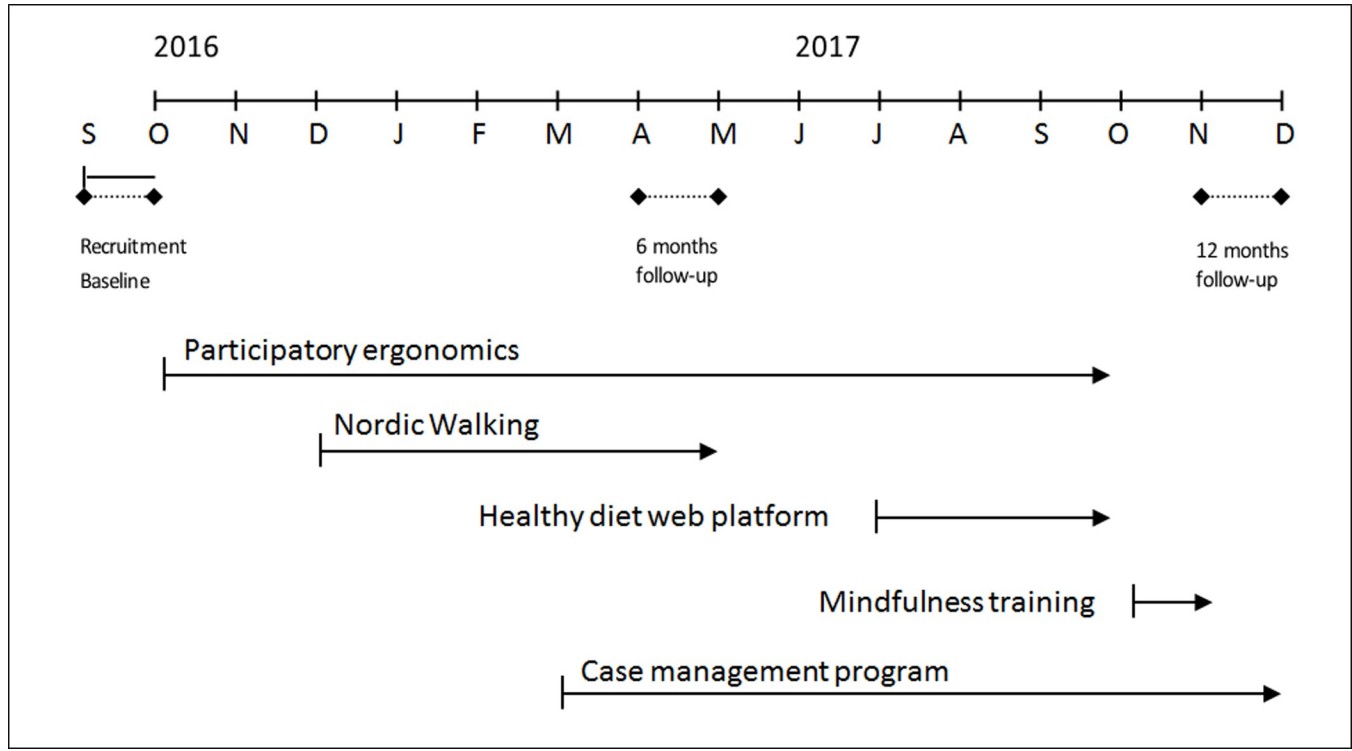

**Fig 2. Timeline of the implementation of the three components for the INTEVAL_Spain intervention.**

of a previously validated self-completed questionnaire that included questions on MSP and exposure to musculoskeletal risk factors at work, and the occupational risk assessment of the unit carried out by the ergonomist. In the treatment phase, the ERGO group was created, composed by the ergonomist of the OHS of the corresponding hospital, a referent worker from each shift (morning, afternoon and two night shifts), the unit supervisor/s, one prevention delegate (union representative) and the project champion. The ERGO group held weekly meetings of one hour each during three weeks, managed by the ergonomist of the OHS. The first meeting consisted of an ergonomics training; at the second meeting, ergonomic problems at the unit were identified and prioritized; and a proposal of preventive measures was developed in the last meeting. In between these meetings, the referents of the ERGO group involved their coworkers in the so-called "Prevention Circles". Prevention circles held discussions in the work units about the subject discussed in the ERGO group meeting, and provided input for the meetings. Finally, the implementation phase consisted of the execution of preventive measures that included organizational, structural, technical, training/information improvements in the workplace. In this phase, the OHS coordinated an "Operative group" composed by the key managers and was coordinated by the head of the OHS with periodical monthly or bimonthly meetings to follow up the process and agreements using a standardized planning table. Moreover, the implementation process was supervised by the Health and Safety Committee of each hospital, the preventive delegates (union representatives), the Chief Executive and the Human Resources Director.

The healthy lifestyle promotion program was developed to encourage healthy lifestyles among workers, including activities such as Nordic walking [37], mindfulness [38] and healthy diet based on the Mediterranean diet [39, 40]. Nordic Walking training consisted of a 12-session program of 1.5 h/session during 12 weeks; Mindfulness was a short course of a weekly

session of 2 hours during 4 weeks, based on Mindfulness-based Stress Reduction (MBRS); the healthy diet component consisted of an online web platform and a three hour chef group session.

*The tailored case management program* for nursing staff with some limitation due to MSP aimed for workers to stay at or return to work in adequate health conditions. The program consisted of early detection of disabling musculoskeletal conditions (MSP and/or MSDs) and support of return to work, through an explicit, exhaustive, multidisciplinary and priority care system [41–43].

**Care as usual.** The OHS carried out the usual occupational health practices for both the intervention and control units. The care as usual included an annual risk assessment, the investigation of all occupational injuries, health surveillance, vaccination, a smoking cessation program, expert assessment in occupational health, a support program for return to work (clinical support, workplace adaptations, management of permanent disability), and training and information in occupational health.

## Primary outcome measures

**Prevalence of self-perceived musculoskeletal pain.** Self-perceived musculoskeletal pain was measured with a validated Spanish adaptation of the Standardized Nordic Questionnaire for the analyses of musculoskeletal symptoms from the ERGOPAR method [30, 44, 45] at baseline, six and 12-month follow-up. Information on pain or discomfort was collected in seven anatomical sites, i.e. (1) neck, shoulders and upper back, (2) low back, (3) elbows, (4) hands, (5) legs, (6) knees and (7) feet. We used the question "do you have pain or discomfort in this anatomical site?" with three possible answers (yes pain, yes discomfort, no pain). For the analysis of each anatomical site, the variable was dichotomized (yes pain or discomfort, no pain) to estimate the improvement of MSP over time, comparing the intervention group with the control group.

**Sickness absence.** Sickness absence was measured as the number of episodes and the number of days on sick leave. The Human Resources Department and the OHS registers provided data on (new) episodes of sickness absence due to musculoskeletal conditions (MSP and/or MSDs) during the study period (October 2016-December 2017) and sickness absence episodes were registered with their ICD-10 diagnostic group code.

## Secondary outcome measure

**Work functioning.** Work functioning was measured at baseline, six and 12 months' follow-up using the Work Role Functioning Questionnaire-Spanish Version (WRFQ-SpV) [14–16]. The WRFQ consists of 27 items divided into five subdomains: work scheduling demands (5 items), output demands (7 items), physical demands (6 items), mental demands (6 items) and social demands (3 items). Each item has five response options ranging from 0 to 4 (0 = difficult all the time (100%), 1 = difficult most of the time, 2 = difficult half of the time (50%), 3 = difficult some of the time, 4 = difficult none of the time (0%)). An additional sixth answer option "not applicable to my work" was also allowed when the work demand was not a part of the job, and was coded as missing. Subscales and the total score used a simple summative approach, multiplying the average of the items by 25 to obtain scores between 0 and 100. High scores indicated better work functioning. If more than 20% of the items were missing, the scale score was set to missing [46].

## Sample size

The sample size calculation was based on the primary outcome MSP, with an estimated prevalence of 80% for healthcare workers [3]. The target of the present study was to reduce the

prevalence of MSP by 20% [47, 48]. We assumed alpha values (type I error) = 0.05 (two-sided), statistical power = 0.80 and an intraclass correlation coefficient (ICC) = 0.05. Using a likelihood-ratio test with these criteria, 82 workers were needed in the intervention and control group, respectively.

## Randomization

Eight clusters were selected, six in one hospital and two in the other hospital. A computerized random allocation stratified by center was developed by an independent researcher to randomize the clusters, obtaining 4 clusters in each group. The intervention group included an intensive care unit (n = 82), a psychogeriatric unit (n = 45), and two traumatology units (n = 39, n = 57). A surgical unit (n = 105), an acute geriatrics unit (n = 34), and two cardiology-respiratory units (n = 39, n = 57) constituted the control group.

## Blinding

This cluster-randomized trial was not blinded. The condition of being in the intervention or the control group could not be blinded, nor could the participating OHS professionals be blinded because they were involved in the implementation of the intervention. The researcher in charge of the analyses was not blinded during the data modeling.

## Pairing data

In our study, data could not be paired as questionnaires were anonymous, and we did not create an identifier. This decision was due to the fact that the questionnaires were self-administered at the workplace and included personal questions, and questions about the relationship with the coworkers and the supervisors. Therefore, the research team decided to administer the questionnaires anonymously and without an identifier so that no one in the institution could identify them, as to encourage participation and make sure that the participants were feeling comfortable with the study.

## Statistical analyses

Baseline descriptive analyses were performed to compare the characteristics of the intervention and control groups using Chi Square Test, or Fisher's exact test when 20% or more cells of the contingency tables had an expected value lower than 5 or one value lower than 1.

The analysis could not take into account intrapersonal variability because data were not paired, and all observations were considered as different individuals and grouped by cluster for analyses. Besides, we could not know who participated in each component of the intervention. Thus, we performed intention-to-treat analyses (ITT) including all workers of analyzed clusters, regardless of whether they had actively participated in the intervention or not. To examine the improvement of MSP over time, from the intervention group with respect to the control group, logistic regression was used through generalized estimation equations (GEE) with an exchangeable correlation structure, taking into account that participants from the same hospital might be correlated. The model was adjusted for occupation, age and baseline prevalence of MSP in each cluster to reduce the effect of the regression to the mean.

The incidence (number of new episodes) and duration (days) of sickness absence were analyzed during the period of the study.

Sickness absence data was available for all workers who agreed to participate in the study and signed the informed consent. Forty-six sickness absence episodes with their ICD-10 diagnostic group code were registered. Of these, 11 were considered as red flags (i.e. bone fractures

and surgery in the previous three months) and excluded from the analyses. Ultimately, recurrent episodes were excluded from the analyses (n = 11). Finally, 24 episodes of sickness absence were included. To estimate the risk of having an episode of sickness absence in the intervention group compared to the control group, a logistic regression model was used. The model was adjusted for occupation, sex and age. Cox proportional hazard model was used to analyze the duration of sickness absence.

Analyses on work functioning compared the improvement over time between the intervention and control group, using a linear regression through GEE with an exchangeable correlation structure, taking into account that participants from the same hospital might be correlated. The model was also adjusted for occupation, sex, age and baseline WRFQ from each cluster.

The analyses were performed with STATA 13 (StataCorp, 2013. Stata Statistical Software: Release 13. College Station, TX: StataCorp LP).

## Results

The eight clusters included 473 nurses and aides. Eighteen people were excluded because they were on sabbatical leave. A total of 164 nurses and aides did not sign formal consent to participate, and 34 did not fill in the baseline questionnaire (Fig 1). During the fieldwork, one of the control clusters, the surgical unit (n = 105), started a complex period of reorganization, did not complete the follow-up questionnaires and left the study.

Of the 257 participants who answered the baseline questionnaire, 138 were in the intervention group and 119 in the control group. The majority of participants were women (84.1%), between 31 and 49 years old, and 63% were nurses. Baseline characteristics were similar between the intervention and the control group (Table 1).

Table 2 shows the participation rates for the different intervention activities. The participation rate of mindfulness was 48.6%, followed by Nordic Walking (36.2%) and participatory ergonomics (21.0%).

### Self-perceived prevalence of MSP

A 63% risk reduction (OR = 0.37; 95% CI = 0.14–0.96) of self-perceived neck, shoulders and upper back pain was observed at 12-month follow-up. The risk of low back pain also decreased though it was not statistically significant. Improvements in other anatomical sites (hands, legs, knees, and feet) were not statistically significant (Table 3).

### Sickness absence incidence

Of 237 workers who agreed to participate in the study and signed the informed consent (excluding the surgical unit), 150 workers belonged to the intervention group and 87 workers to the control group. A total of 24 new, non-recurrent episodes of sickness absence related to musculoskeletal conditions were registered (Table 4).

The incidence of sickness absence was slightly higher in the intervention group compared to the control group, but no statistically significant differences were found regarding the incidence or the days of sickness absence until return to work for intervention and control group.

### Work functioning

Work functioning remained stable with a total score around 80 in both the intervention and the control group, i.e. participants with a full-time work week experienced difficulties in performing their work demands 1 day a week (Table 5). Work scheduling demands showed a

**Table 1. Baseline characteristics of workers by study group (intervention and control group).**

| Characteristics | | Total (n = 257) | | Intervention (n = 138) | | Control (n = 119) | | |
|---|---|---|---|---|---|---|---|---|
| | | n | % | n | % | n | % | p |
| **Sex** | | | | | | | | |
| | Male | 30 | 11.7 | 16 | 11.6 | 14 | 11.8 | 0.852[1] |
| | Female | 216 | 84.1 | 117 | 84.8 | 99 | 83.2 | |
| | *missing* | 11 | 4.2 | 5 | 3.6 | 6 | 5.0 | |
| **Age (years)** | | | | | | | | |
| | ≤30 | 41 | 16.0 | 19 | 13.8 | 22 | 18.5 | 0.145[1] |
| | 31–49 | 136 | 52.9 | 69 | 50.0 | 67 | 56.3 | |
| | ≥49 | 80 | 31.1 | 50 | 36.2 | 30 | 25.2 | |
| **Shift** | | | | | | | | |
| | Morning | 99 | 38.5 | 53 | 38.4 | 46 | 38.7 | 0.491[1] |
| | Afternoon | 64 | 25.0 | 30 | 21.7 | 34 | 28.6 | |
| | Night | 75 | 29.2 | 44 | 31.9 | 31 | 26.1 | |
| | Other | 14 | 5.5 | 7 | 5.1 | 7 | 5.9 | |
| | *missing* | 5 | 2.0 | 4 | 2.9 | 1 | 0.8 | |
| **Contract** | | | | | | | | |
| | Official | 8 | 3.1 | 7 | 5.1 | 1 | 0.8 | 0.099[2] |
| | Indefinite | 159 | 61.9 | 85 | 61.6 | 74 | 62.2 | |
| | Interim | 37 | 14.4 | 21 | 15.2 | 16 | 13.5 | |
| | Temporal | 43 | 16.7 | 21 | 15.2 | 22 | 18.5 | |
| | Other | 4 | 1.6 | 0 | 0.0 | 4 | 3.4 | |
| | *missing* | 6 | 2.3 | 4 | 2.9 | 2 | 1.7 | |
| **Occupation** | | | | | | | | |
| | Nurse | 161 | 62.7 | 78 | 56.5 | 83 | 69.8 | 0.069[1] |
| | Nursing aide | 95 | 37.0 | 59 | 42.8 | 36 | 30.3 | |
| | *missing* | 1 | 0.4 | 1 | 0.7 | 0 | 0.0 | |

[1]Chi-Square Test;

[2]Fisher's Exact Test.

statistically significant decrease between baseline and 12 months follow-up in the intervention compared to the control group (p = 0.043). No other domains of work functioning showed statistically significant differences between the baseline and at 12 months follow-up in the intervention compared to the control group.

**Table 2. Participation rates of intervention activities.**

| Activities participation | Intervention (n = 138) | |
|---|---|---|
| | n | % |
| Participatory Ergonomics | 29[1] | 21.0 |
| Case Management | 13 | 9.4 |
| Nordic Walking | 50 | 36.2 |
| Mindfulness | 67 | 48.6 |
| Healthy Diet | 7[2] | 5.1 |

[1]Number of workers directly involved in the ERGO Group.

[2]Number of workers that attended.

**Table 3. Prevalence of self-perceived MSP in the intervention group compared to the control.**

| Self-perceived pain | Intervention group | | | | | | Control group | | | | | | I-C | | I-C | |
|---|---|---|---|---|---|---|---|---|---|---|---|---|---|---|---|---|
| | Baseline (n = 138) | | 6 months (n = 125) | | 12 months (n = 105) | | Baseline (n = 119) | | 6 months (n = 63) | | 12 months (n = 79) | | Baseline-6 months | | Baseline- 12 months | |
| | n | % | n | % | n | % | n | % | n | % | n | % | OR | 95% CI | OR | 95% CI |
| Neck/upper back[1] | 120 | 87.0 | 100 | 80.0 | 71 | 67.6 | 87 | 73.1 | 45 | 71.4 | 56 | 73.7 | 1.04 | 0.38–2.82 | 0.37* | 0.14–0.96 |
| Low back | 107 | 77.5 | 88 | 70.0 | 58 | 55.2 | 93 | 78.2 | 48 | 76.2 | 36 | 47.4 | 1.34 | 0.49–3.65 | 0.65 | 0.25–1.68 |
| Elbows | 13 | 9.4 | 16 | 12.8 | 7 | 6.7 | 13 | 10.9 | 4 | 6.4 | 2 | 2.6 | 3.59 | 0.89–1.45 | 2.02 | 0.42–9.85 |
| Hands | 46 | 33.3 | 40 | 32.0 | 27 | 25.7 | 28 | 23.5 | 20 | 31.8 | 13 | 17.1 | 0.85 | 0.35–2.06 | 0.92 | 0.36–2.30 |
| Legs | 66 | 47.8 | 42 | 33.6 | 30 | 28.6 | 48 | 40.3 | 26 | 41.3 | 21 | 27.6 | 0.54 | 0.23–1.23 | 0.58 | 0.25–1.34 |
| Knees | 40 | 29.0 | 36 | 28.8 | 24 | 22.9 | 32 | 26.9 | 21 | 33.3 | 15 | 19.7 | 0.89 | 0.37–2.09 | 0.74 | 0.31–1.79 |
| Feet | 50 | 36.2 | 36 | 28.8 | 25 | 23.8 | 36 | 30.2 | 21 | 33.3 | 17 | 23.4 | 0.72 | 0.31–1.68 | 0.52 | 0.22–1.23 |

*p<0.05.

[1]Neck, shoulders and upper back.

I: Intervention; C: Control. GEE model adjusted by baseline, age and occupation. Reference categories were: time at baseline and control group.

## Discussion

This cluster randomized controlled trial evaluating the effectiveness of the multifaceted intervention INTEVAL_Spain that includes participatory ergonomics, health promotion and case management components to reduce and manage MSP was effective in reducing neck, shoulders and upper back pain in hospital nursing staff exposed to high physical demands. A non-statistically significant reduction of low back pain was shown in the intervention group as compared to the control clusters. No reduction of other pain locations nor of the incidence and duration of related sickness absence was observed. Work functioning remained stable along the 12 months of follow-up.

The reduction of the risk of neck, shoulders and upper back pain supports our hypothesis that a multifaceted intervention encompassing the three levels of prevention might be effective to prevent MSP. This result also agrees with the effectiveness observed for other multifaceted interventions and studies that incorporate participatory ergonomics, with an improvement of MSP in workers, including nursing staff [19, 44, 48]. The lack of statistical significance on the observed risk reduction for low back pain could be due to the number of participants responding to the questionnaires that could have had limited the power of the study. Our results do not show any other statistically significant improvements of MSP in other sites (i.e. elbows,

**Table 4. Incidence of sickness absence and days until return to work in the intervention group compared with the control group.**

| Sickness absence | Total (n = 237)[3] | | Intervention (n = 150) | | Control (n = 87) | | Regression analyses | |
|---|---|---|---|---|---|---|---|---|
| Sickness absence during the study | | | | | | | | |
| | n | % | n | % | n | % | OR | 95% CI |
| Incidence[1] (n° episodes, %) | 24 | 10.1 | 18 | 12.0 | 6 | 6.9 | 1.64 | 0.63–4.28 |
| | | | | | | | HR | 95% CI |
| Duration[2] MD, IQR | 25 | (12.8–91) | 28.5 | (13.3–114) | 21 | (10.5–67) | .57 | 0.20–1.67 |

[1] Logistic regression model

[2] Cox regression model

[3] Total number of workers who agreed to participate in the study (signed consents), excluding surgical unit.

Models adjusted by occupation, sex and age. MD, median; IQR, IQR 25th-75th percentile.

**Table 5. Differences in work functioning in the intervention group compared to the control group.**

| | INTERVENTION GROUP | | | | | | CONTROL GROUP | | | | | | TOTAL |
|---|---|---|---|---|---|---|---|---|---|---|---|---|---|
| | Baseline (n = 138) | | 6 months (n = 125) | | 12 months (n = 105) | | Δ | Baseline (n = 119) | | 6 months (n = 63) | | 12 months (n = 79) | | Δ | Δ I-C |
| | m | 95% CI | m | 95% CI | m | 95% CI | | m | 95% CI | m | 95% CI | m | 95% CI | | |
| WS | 87.2 | 84.4–90.0 | 86.1 | 83.2–89.0 | 82.2 | 79.0–85.4 | 5 | 90.3 | 87.4–93.3 | 86.5 | 82.5–90.4 | 91.3 | 87.7–95.0 | -1 | 6.0[*] |
| OD | 82.3 | 79.4–85.1 | 81.2 | 78.3–84.1 | 79.2 | 75.9–82.4 | 3.1 | 84.4 | 81.4–87.5 | 81.6 | 77.5–85.7 | 83.3 | 79.6–87.1 | 1.1 | 2.0 |
| PD | 81.4 | 77.2–85.5 | 80.2 | 75.9–84.4 | 78.5 | 73.9–83.1 | 2.9 | 84.3 | 80.0–88.7 | 82.3 | 76.8–87.7 | 85.4 | 80.2–90.7 | -1.1 | 4.0 |
| MD | 88.2 | 85.5–90.9 | 86.4 | 83.6–89.2 | 90.2 | 87.1–93.2 | -2 | 88.8 | 85.9–91.6 | 87.2 | 83.4–91.0 | 89.6 | 86.1–93.1 | -0.8 | -1.2 |
| SD | 90.0 | 87.8–92.1 | 89.7 | 87.4–91.9 | 92.9 | 90.3–95.5 | -2.9 | 90.6 | 88.2–93.0 | 91.1 | 87.8–94.3 | 94.0 | 91.0–97.0 | -3.4 | 0.5 |
| OS | 85.0 | 82.7–87.3 | 83.9 | 81.5–86.3 | 83.3 | 80.7–86.0 | 1.7 | 87.2 | 84.7–89.7 | 85.0 | 81.7–88.3 | 87.7 | 84.7–90.7 | -0.5 | 2.2 |

[*] p-value < 0.05.

m: mean. Δ: Baseline-12 months differences. WS: work scheduling demands; OD: output demands; PD; physical demands; MD: mental demands; SD: social demands; OS: overall score. I: Intervention; C: Control. GEE model adjusted by baseline, age and occupation.

hands, legs, knees and feet) although there was a decreasing trend of pain in the intervention group at follow-up. It may be that the design of the study favored the most prevalent sites. Regarding participatory ergonomics workers evaluated their risk factors and prioritized the identified preventive measures, so it is possible that they focused on the most prevalent pain locations, i.e. neck, shoulders, upper back and low back. Therefore, it may be that they proposed fewer measures to reduce pain in the less prevalent sites, i.e. hands, elbows, legs, knees and feet. Actually, most studies tend to focus only on low back pain [5, 7, 8, 19, 22, 40, 42] as is the most prevalent site [1, 2]. Another explanation could be that the least frequent pain locations are less related to ergonomic and others exposures at work in nursing staff. Preventive measures included structural, technical, organizational, training/information improvements in the workplace. Although the majority of proposed measures were implemented, It was not feasible to implement relevant other measures, especially the most expensive measures (e.g. staff recruitment, changing the rooms structure, etc.) and those that involved an expansion of the workforce. Perhaps these limitations explain the smaller than expected impact.

No significant differences were found in the incidence or duration of sickness absence in the intervention compared to the control group. Several studies of multifaceted interventions on MSP also include sickness absence, due to the relationship between MSP as a risk factor for sickness absence [7, 18, 19, 49]. One of the components of our intervention was case management which has shown its effectiveness to reduce the duration of sickness absence, musculoskeletal symptoms and disability, and improvements of work continuity in several studies from different countries [42, 43, 50]. However, the evidence is conflicting about the effectiveness of multifaceted interventions to reduce sickness absence [42, 51], or it may be that this lack of effect could be due to methodological limitations [19]. In our study, case management was a tailored service, so the participation in each component of the intervention was not equal. All workers in each hospital unit may have benefited from participatory ergonomics, but only those who had a limiting MSP or sickness absence actually participated in case management with a small impact at cluster level. Otherwise, case management in occupational health is not an extensive practice in Spain so it was a very innovative process in these hospitals, and it could be difficult to integrate it and get the understanding by workers and their managers.

MSP may have a high impact to maintain functioning at work, and quality of life of nurses [19]. Work functioning remained stable around 80% in both the intervention and control groups, except in work scheduling demands that showed a statistically significant decrease in

in the intervention group compared to the control group. Despite this, the score was still above 80%. This score has been reported by other authors [16] in the actively working population, which means that the worker is able to perform his/her work tasks in 80% of the time. More-over, the intervention was not directed towards the improvement of work functioning and studies are needed that address the interplay of work and health in work functioning.

Finally, the use of multifaceted interventions is highly recommended by other authors [5, 18, 19, 21] for two main reasons: the multi-causality of MSP that demands a biopsychosocial approach [24–26] and the increased possibility to allow workers to participate in a compo-nent which fits their needs [19]. We believe that our intervention offers this biopsychosocial approach, since it includes the three components that encompass the different dimensions of health, i.e. physical, psychological, occupational and social environments, allowing each nurse and nursing aide to choose participation in the best component that met his/her needs.

A strength of this study is the cluster randomized controlled trial design, which is consid-ered the methodological paradigm for the evaluation of health interventions since it ensures that other factors that may be related to the MSP are distributed randomly [47]. Moreover, the intervention was designed to achieve a biopsychosocial management of MSP, encom-passing the three levels of prevention, and including components based on the scientific evidence.

The main limitation of this study is that the data were not paired and consequently the analysis could not take into account intrapersonal variability, which adds difficulties in finding a significant effect of the intervention. Also, the final number of participants answering the questionnaires and the loss of one cluster during the follow-up could have had an impact on the power of the study. To some extent our study was a black box as we analyzed the intervention components and outcomes without evaluating the intermediate mechanisms between the intervention components and the MSP and sickness absence out-comes beyond the scope of our study. We do not know the effect of each component of the intervention separately, nor if there could be a gradient between MSP and the number of components. However, a process evaluation is ongoing. We had no information about the services that participants of the intervention or control groups received outside the study, as it might be that participants in the control group visited physiotherapists, or joined a Nordic walking group out of work, which could had been a source of contamination in our study. Finally, even though the clusters were different, and independent units were located on dif-ferent floors and sometimes centers of the hospital, contamination cannot be ruled out, which may have resulted in an underestimation of the evaluated effects.

We recommend that future research focus on the examination of the specific and interme-diate mechanisms between the components of the intervention in the reduction of MSP and on the core components, to understand the full process from the intervention to the outcomes. Also, we recommend including a follow-up after the end of the intervention and develop this intervention in other occupational groups. Strategies that focus on reducing MSP in other ana-tomical sites would therefore be required.

## Conclusion

There is a need to develop multifaceted interventions focused on preventing and managing MSP in nursing staff and other occupational groups. Our intervention showed to be effective for reducing neck, shoulder and upper back pain. Our results, though modest, suggest that interventions to reduce and manage MSP need a multifactorial approach including the three levels of prevention.

## Supporting information

**S1 File. Checklist.** CONSORT 2010 checklist of information to include when reporting a randomised trial.
(DOC)

**S2 File. Study protocol (Spanish version).** Evaluación de una intervención multifacética para reducir y gestionar el dolor musculoesquelético en personal de enfermería (INTEVAL_Spain).
(PDF)

**S3 File. Study protocol (English version).** Evaluation of a multifaceted intervention to reduce and manage musculoskeletal pain in nurses (INTEVAL_Spain).
(PDF)

**S4 File. Database.** Musculoskeletal Pain and Work Role Functioning Questionnaire INTEVAL_Spain database.
(XLS)

## Acknowledgments

The members of INTEVAL_Spain research team are José Maria Ramada, Consol Serra, Mercè Soler-Font, Antoni Merelles (Nursing Department, Nursing and Podiatry Faculty, University of Valencia, Valencia, Spain); Pilar Peña (Occupational Health Service, Corporació Sanitària Parc Taulí, Sabadell, Spain); and, Sergio Vargas-Prada (Healthy Working Lives Group, Institute of Health and Wellbeing, College of Medical, Veterinary and Life Sciences, University of Glasgow, Glasgow, UK).

We want to thank all healthcare workers and their representatives, referents, managers and supervisors from the hospital clusters of Parc de Salut Mar (PSMar) and Corporació Sanitaria Parc Taulí (CSPT) who agreed to participate in the trial. Especially, PSMar: Pilar Pastor (Ward manager), Isabel Aranega, Noemí Cajete, Raúl Martín, Dolores Rincón, Nuria Saavedra (UH30);

Rosa Balaguer (ward manager), Sonia Advíncula, Nuria Esteban, Montse Regordosa, Cristina Salvat, Ana Uribe (Intensive Care Unit); Isabel Egea (ward manager), Ana Delgado, M Ángeles Fernández, Josefa García, Susana Margalef, Alexandra Morales, Ana M Rodríguez, Isabel Rodríguez (Llevants 3,4); Montse Sitges, Txell Gumà (ward managers), Rosa Elias, Lucía Fernández, Ana M Luque, Nuria Morillas, Carlos Perez, Sandra Vives (Surgical area); Elena Maull, Desirée Ruiz (ward manager), Alberto Gonzalez, Antonia Rincón, Bernat Sarrió, Gina Shakya (UH04); Beatriz Fernández (ward manager), Mª Encarnación Avilés, Miriam Hernández, Naza Martinez, Carme Pellín, Nenita de los Santos, Sergio Taibo, Chari Villanueva (Acute geriatrics unit). Rosa Aceña, Cuca Esperanza and Núria Pujolar (Nursing Coordinators); Mercedes Calvo, Miguel Celada, Lluisa Cosp, Eugenio Gurrea, Montse Sallés, Pilar Serrano (Nurse Supervisors); and Vicky Abad, Pilar González, Francisco Martos (Prevention Delegates). CSPT: Isabel Simó (ward manager), Mª Goretti Gelonch, Elisabeth Mérida, Sara Purcalla, Mónica Sianes (UH06); José Mª Barradas (ward manager), Judith Camps, M José González, Verònica Gómez, Victoria Plaza, Estefanía del Pino (UH08); and Elena Polo (Prevention Delegate).

We also want to thank the contribution of Chelo Sancho (specialist in participatory ergonomics), Rocío Villar (occupational physician, PSMar); Victòria Lopez (occupational nurse, CSPT); Cristina Cervantes, Ferran Escalada and the physiotherapists team (Rehabilitation Service, PSMar); Fernanda Caballero and physiotherapists team (Rehabilitation Service, CSPT); Gemma Salvador (Agència de Salut Pública de Catalunya) and Ada Parellada (Chef); Anna

Amat (champion and case manager), Carmen de la Flor (champion), Montserrat Fernandez (CiSAL, UPF); Cristina Giménez (Psychologist); Antonio Brieba (Nordic walking instructor); Georgina Badosa and Mónica Astals (Mindfulness instructors); and Monica Ubalde-Lopez for her contributions during the drafting of the manuscript and Andrew N March for providing a grammatical revision of the text.

## Author Contributions

**Conceptualization:** Mercè Soler-Font, José Maria Ramada, Consol Serra.

**Data curation:** Mercè Soler-Font, Josué Almansa.

**Formal analysis:** Mercè Soler-Font, Sander K. R. van Zon, Josué Almansa.

**Funding acquisition:** José Maria Ramada, Consol Serra.

**Investigation:** Mercè Soler-Font, José Maria Ramada, Consol Serra.

**Methodology:** Mercè Soler-Font, José Maria Ramada, Sander K. R. van Zon, Ute Bültmann, Consol Serra.

**Project administration:** Consol Serra.

**Resources:** Consol Serra.

**Supervision:** José Maria Ramada, Sander K. R. van Zon, Ute Bültmann, Consol Serra.

**Writing – original draft:** Mercè Soler-Font.

**Writing – review & editing:** José Maria Ramada, Sander K. R. van Zon, Josué Almansa, Ute Bültmann, Consol Serra.

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
