## [Decision Letter · Decision Letter 0]

15 Aug 2019

PONE-D-19-15120

Multifaceted intervention for the prevention and management of musculoskeletal pain in nursing staff: results of a cluster randomized controlled trial

PLOS ONE

Dear Dr Serra,

Thank you for submitting your manuscript to PLOS ONE. After careful consideration, we feel that it has merit but does not fully meet PLOS ONE’s publication criteria as it currently stands. Therefore, we invite you to submit a revised version of the manuscript that addresses the points raised during the review process.Consider in particular the methodological comments of reviewer 1 and provide full availability of data. Consider also the other points raised by the second reviewer.

We would appreciate receiving your revised manuscript by September 20. To enhance the reproducibility of your results, we recommend that if applicable you deposit your laboratory protocols in protocols.io, where a protocol can be assigned its own identifier (DOI) such that it can be cited independently in the future. For instructions see: http://journals.plos.org/plosone/s/submission-guidelines#loc-laboratory-protocols

We look forward to receiving your revised manuscript.

Kind regards,

Andrea Martinuzzi

Academic Editor

PLOS ONE

2. Thank you for submitting your clinical trial to PLOS ONE and for providing the name of the registry and the registration number. The information in the registry entry suggests that your trial was registered after patient recruitment began. PLOS ONE strongly encourages authors to register all trials before recruiting the first participant in a study.

1) your reasons for your delay in registering this study (after enrolment of participants started);

2) confirmation that all related trials are registered by stating: “The authors confirm that all ongoing and related trials for this drug/intervention are registered”.

3) an explanation on any discrepancy between the text and the protocol (for example, we note that sample size calculation slightly differs between protocol and Methods section)

Please also ensure you report the date at which the ethics committee approved the study as well as the complete date range for patient recruitment and follow-up in the Methods section of your manuscript.

4. We note you have included a table to which you do not refer in the text of your manuscript. Please ensure that you refer to Table 3 in your text; if accepted, production will need this reference to link the reader to the Table.

Reviewers' comments:

Reviewer's Responses to Questions

**Comments to the Author**

1. Is the manuscript technically sound, and do the data support the conclusions?

Reviewer #1: Partly

Reviewer #2: Yes

2. Has the statistical analysis been performed appropriately and rigorously? 

Reviewer #1: Yes

Reviewer #2: Yes

3. Have the authors made all data underlying the findings in their manuscript fully available?

Reviewer #1: No

Reviewer #2: No

4. Is the manuscript presented in an intelligible fashion and written in standard English?

Reviewer #1: Yes

Reviewer #2: Yes

5. Review Comments to the Author

Reviewer #1: This paper describes an interesting study that was well designed with an impressive sample size and a long followup period. However there are some major issues to be addressed specifically:

1. The Interventions involved primary measures that were described as "participatory ergonomics" - which involved the workers to identify their occupational risk factors. However - there is no specific description of what this process involves. Some specific details were found in the "study protocol" that was published in Serra et al, BMC Public Health 2018, 19, 348. It still needs to be clearly explained, for example, "who" were the persons to conduct the participatory ergonomics? how many sessions were involved? what were the actual changes in the workplace or work practice that resulted from this process? were these interventions standardised in the different units?

If this is a RCT, the authors should follow the CONSORT guidelines in reporting the interventions which required details such as who provided the interventions and how they were provided. The information in the Appendix was a "proposal" and it did not indicate whether such a protocol was actually implemented. This information should be given within the main text.

2. The interventions such as Nordic walking and "mindfulness" training, how often were these carried out? What was the compliance rate among the participants?

3. To be effective in implementing ergonomic interventions, it requires the cooperation of employers. Were there any actual changes in the work organisation or work practice among the nurses after identifying the occupational risk factors?

4. If the process does not involve any real changes in work habits or work organisation, it may be the reason why there was no significant changes in the results produced.

5. In reporting the musculoskeletal pain, the "prevalence" rates were compared in terms of %. Is this just a "yes/no" answer in the questionnaire? Did you compare the pain rating for different body areas?

Reviewer #2: A two arm randomized cluster controlled trial was conducted to evaluate an intervention to prevent or manage musculoskeletal pain in nurses. Compared to the control group, a decreased risk in shoulder, neck and back pain was observed in the intervention group. The manuscript is clearly written.

Minor revisions:

1- Abstract: Capitalize Cox.

2 Line 193: Indicate the statistical testing method which achieves an 80% power and state if the alpha level is one- or two-sided.

3- Table 1: State the statistical testing methods used to estimate the p-values.

6. PLOS authors have the option to publish the peer review history of their article (what does this mean?). If published, this will include your full peer review and any attached files.

Reviewer #1: No

Reviewer #2: No

---

## [Author Response · Author response to Decision Letter 0]

27 Sep 2019

Dr Andrea Martinuzzi,

Academic Editor of “PLOS ONE”,

Subject: Resubmission of revised manuscript [Manuscript ID PONE-D-19-15120] - “Multifaceted intervention for the prevention and management of musculoskeletal pain in nursing staff: results of a cluster randomized controlled trial”. 

Dear Editor, 

We thank the Editor and the reviewers for their thoughtful and constructive comments to improve our manuscript, entitled ““Multifaceted intervention for the prevention and management of musculoskeletal pain in nursing staff: results of a cluster randomized controlled trial”. In the enclosed response, we have addressed all reviewer’s and editorial comments. The most relevant additions in the article are: 1) information about the specific procedure of the participatory ergonomics, Nordic Walking and mindfulness components of the intervention, 2) information on workers’ participation in the intervention activities, and 3) information on the complete date range for patient recruitment and follow-up. We trust you will find that this revision addresses all raised concerns and we appreciate your re-consideration of our manuscript for publication in PLOS ONE.

Moreover, we have submitted the dataset with the descriptive variables, and the outcomes of musculoskeletal pain and the Work Role Functioning Questionnaire (WRFQ) as a supporting information file.

Yours sincerely,

also on behalf of the co-authors (Mercè Soler-Font, José Maria Ramada, Sander K.R. van Zon, Josué Almansa, Ute Bültmann),

Consol Serra,

Corresponding author

 

JOURNAL REQUIREMENTS:

REQUIREMENT 1: Please ensure that your manuscript meets PLOS ONE's style requirements, including those for file naming. 

RESPONSE: We have reviewed the PLOS ONE’s style requirements, and modified the corresponding author information. Also, we have moved the section of “on behalf of INTEVAL_Spain research team” to the Acknowledgements.

REQUIREMENT 2: Thank you for submitting your clinical trial to PLOS ONE and for providing the name of the registry and the registration number. The information in the registry entry suggests that your trial was registered after patient recruitment began. PLOS ONE strongly encourages authors to register all trials before recruiting the first participant in a study. As per the journal’s editorial policy, please include in the Methods section of your paper:

1) Your reasons for your delay in registering this study (after enrolment of participants started).

2) Confirmation that all related trials are registered by stating: “The authors confirm that all ongoing and related trials for this drug/intervention are registered”. 

3) An explanation on any discrepancy between the text and the protocol (for example, we note that sample size calculation slightly differs between protocol and Methods section)

RESPONSE: 

We have included all these specifications in the Methods section. 

1) This trial was retrospectively registered because it was not a requirement for the approval of the protocol by the Ethical Committee of Parc de Salut Mar. 

2) We have added the sentence “The authors confirm that all ongoing and related trials for this drug/intervention are registered” (page 6, paragraph 5, lines 119-120). 

3) We have repeated the sample size calculation. The values in this manuscript and in the “protocol study” article are correct. We confirm that the sample size calculation slightly differs between the protocol and the manuscript due to a type error in the protocol. Another discrepancy between the text and the protocol concerns the secondary outcome Organizational Preventive Culture as this could not be analyzed because of missing data. 

[Page 6, paragraph 5, lines 119-124]

The authors confirm that all ongoing and related trials for this intervention are registered. INTEVAL_Spain was retrospectively registered in ISRCTN (registration number 15780649). This trial was retrospectively registered because it was not a requirement for the approval of the protocol by the Ethical Committee of Parc de Salut Mar. A discrepancy between the text and the protocol concerns the secondary outcome Organizational Preventive Culture as this could not be analyzed because of missing data. 

REQUIREMENT 3: Please also ensure you report the date at which the ethics committee approved the study as well as the complete date range for patient recruitment and follow-up in the Methods section of your manuscript.

RESPONSE: We have added the date that the ethics committee approved the study in the sentence “The Clinical Research Ethical Committee of Parc de Salut Mar provided the ethical approval of the study on July 10, 2014” [Page 6, paragraph 5, lines 118-119].

Also, we have modified Figure 1 to incorporate the complete date range for the patient recruitment and follow-up, and we have specified this information in the Methods section. The revised text reads:

[Page 7, paragraph 2, lines 132-140]

The entire intervention lasted one year, starting at the beginning of September 2016. During the first month, the participant’s recruitment was carried out and the baseline questionnaire was completed. After cluster randomization, the intervention started a month later with participatory ergonomics as this intervention required a longer process to be fully implemented, e.g. purchase of materials. Case management was carried out from March 2017 to December 2017. The healthy lifestyle promotion program started in December 2016 with Nordic Walking, a healthy diet web platform and mindfulness training. Follow-up questionnaires were administered at 6 and 12 months follow-up, during April and November 2017.

 [Figure 1]

Figure 1: Timeline of the implementation of the three components for the INTEVAL_Spain intervention.

REQUIREMENT 4: In your Data Availability statement, you have not specified where the minimal data set underlying the results described in your manuscript can be found. PLOS defines a study's minimal data set as the underlying data used to reach the conclusions drawn in the manuscript and any additional data required to replicate the reported study findings in their entirety. All PLOS journals require that the minimal data set be made fully available. For more information about our data policy, please see http://journals.plos.org/plosone/s/data-availability. Upon re-submitting your revised manuscript, please upload your study’s minimal underlying data set as either Supporting Information files or to a stable, public repository and include the relevant URLs, DOIs, or accession numbers within your revised cover letter. For a list of acceptable repositories, please see http://journals.plos.org/plosone/s/data-availability#loc-recommended-repositories. Any potentially identifying patient information must be fully anonymized. Important: If there are ethical or legal restrictions to sharing your data publicly, please explain these restrictions in detail. Please see our guidelines for more information on what we consider unacceptable restrictions to publicly sharing data: http://journals.plos.org/plosone/s/data-availability#loc-unacceptable-data-access-restrictions. Note that it is not acceptable for the authors to be the sole named individuals responsible for ensuring data access. We will update your Data Availability statement to reflect the information you provide in your cover letter.

RESPONSE: We have submitted the dataset with the descriptive variables, and the outcomes of musculoskeletal pain and Work Role Functioning Questionnaire (WRFQ) as a Supporting Information file.

REQUIREMENT 5: We note you have included a table to which you do not refer in the text of your manuscript. Please ensure that you refer to Table 3 in your text; if accepted, production will need this reference to link the reader to the Table.

RESPONSE: We thank the editor for noting this. We now have referred to Table 3 (Table 4 in the revised version) in the text.

[Page 15, paragraph 2, lines 312-313]

A total of 24 new, non-recurrent episodes of sickness absence related to musculoskeletal conditions were registered (Table 4).

COMMENTS TO THE AUTHOR

REVIEWER 1: This paper describes an interesting study that was well designed with an impressive sample size and a long follow-up period. However there are some major issues to be addressed specifically:

COMMENT 1: The Interventions involved primary measures that were described as "participatory ergonomics" - which involved the workers to identify their occupational risk factors. However - there is no specific description of what this process involves. Some specific details were found in the "study protocol" that was published in Serra et al, BMC Public Health 2018, 19, 348. It still needs to be clearly explained, for example, "who" were the persons to conduct the participatory ergonomics? How many sessions were involved? What were the actual changes in the workplace or work practice that resulted from this process? Were these interventions standardized in the different units? If this is a RCT, the authors should follow the CONSORT guidelines in reporting the interventions which required details such as who provided the interventions and how they were provided. The information in the Appendix was a "proposal" and it did not indicate whether such a protocol was actually implemented. This information should be given within the main text.

RESPONSE: We thank the reviewer for these comments. We indeed used the CONSORT statement to describe our study but we agree with the reviewer that we could have been clearer regarding the description of the interventions. A more detailed description of the participatory ergonomics component of the intervention has been added to the Methods Section. The participatory ergonomics component followed a standardized procedure named ERGOPAR. The ergonomist in charge was the same for all units at each hospital and followed the above cited standardized procedure. 

[Page 8 paragraph 1, lines 148-169]

Participatory ergonomics was divided into three phases: diagnostic, treatment and implementation. It began with the diagnostic phase consisting of the administration of a previously validated self-completed questionnaire that included questions on MSP and exposure to musculoskeletal risk factors at work, and the occupational risk assessment of the unit carried out by the ergonomist. In the treatment phase, the ERGO group was created, composed by the ergonomist of the OHS of the corresponding hospital, a referent worker from each shift (morning, afternoon and two night shifts), the unit supervisor/s, one prevention delegate (union representative) and the project champion. The ERGO group held weekly meetings of one hour each during three weeks, managed by the ergonomist of the OHS. The first meeting consisted of an ergonomics training; at the second meeting, ergonomic problems at the unit were identified and prioritized; and a proposal of preventive measures was developed in the last meeting. In between these meetings, the referents of the ERGO group involved their coworkers in the so-called “Prevention Circles”. Prevention circles held discussions in the work units about the subject discussed in the ERGO group meeting, and provided input for the meetings. Finally, the implementation phase consisted of the execution of preventive measures that included organizational, structural, technical, training/information improvements in the workplace. In this phase, the OHS coordinated an “Operative group” composed by the key managers and was coordinated by the head of the OHS with periodical monthly or bimonthly meetings to follow up the process and agreements using a standardized planning table. Moreover, the implementation process was supervised by the Health and Safety Committee of each hospital, the preventive delegates (union representatives), the Chief Executive and the Human Resources Director.

 [Page 5, paragraph 4, lines 94-96]

The CONSORT statement and the CONSORT extension for cluster randomized trials were used to describe the study design [28, 29]. More detailed information on the content of the intervention and the evaluation process has been described elsewhere [27]. 

COMMENT 2. The interventions such as Nordic walking and "mindfulness" training, how often were these carried out? What was the compliance rate among the participants?

RESPONSE: The mindfulness training was a short course based on the Mindfulness-based Stress Reduction (MBRS) training and consisted of a weekly session of 2 hours during 4 weeks. Nordic Walking training consisted of a 12-session program of 1.5 h/session during 12 weeks by accredited instructors. We have added this information to the Methods section.

[Page 9, paragraph 1, 172-175]

Nordic Walking training consisted of a 12-session program of 1.5 h/session during 12 weeks; Mindfulness was a short course of a weekly session of 2 hours during 4 weeks, based on Mindfulness-based Stress Reduction (MBRS); the healthy diet component consisted of an online web platform and a three hour chef group session.

We have added information on the participation rates to Table 2 and in the text. 

 [Page 14, paragraph 1, lines 292-294]

Table 2 shows the participation rates for the different intervention activities. The participation rate of mindfulness was 48.6%, followed by Nordic Walking (36.2%) and participatory ergonomics (21.0%). 

[Page 14, Table 2, lines 295-297]

Table 2. Participation rates of intervention activities

Activities participation Intervention (n=138)

 n %

Participatory Ergonomics 291 21.0

Case Management 13 9.4

Nordic Walking 50 36.2

Mindfulness 67 48.6

Healthy Diet 72 5.1

1Number of workers directly involved in the ERGO Group.

2Number of workers that attended

COMMENT 3. To be effective in implementing ergonomic interventions, it requires the cooperation of employers. Were there any actual changes in the work organization or work practice among the nurses after identifying the occupational risk factors? 

RESPONSE: Actually, employers’ involvement was a key factor to carry out the ergonomic intervention. Before the intervention, an agreement to fully implement it was signed by with the Chief Executive and the Director of Human Resources in both institutions. After the ERGO group identified occupational risk factors and preventive measures, an “Operative group” was established for the planning and implementation of proposed improvements. It included key managers within the organization from Human Resources, Nurses Management, General Services Department, etc. and was coordinated by the head of the OHS. The implementation of preventive measures included improvements on the work organization and of work practice, such as having reinforcement personnel in shifts with more physical load, reassignment of tasks especially those related to handling of highly dependent patients, improvement of the organization related to hospital discharge with an impact on the burden of nursing personnel, as well as structural, technical or training/information changes. The implementation process was supervised by the Health and Safety Committee of each hospital to ensure the Chief Executive and the Human Resources Director. A budget to cover specific measures was ensured by the top management and coordinated through the occupational health service. 

[Page 8, paragraph 1, line 161-167]

Finally, the implementation phase consisted of the execution of preventive measures that included organizational, structural, technical, training/information improvements in the workplace. In this phase, the OHS coordinated an “Operative group” composed by the key managers and was coordinated by the head of the OHS with periodical monthly or bimonthly meetings to follow up the process and agreements using a standardized planning table. Moreover, the implementation process was supervised by the Health and Safety Committee of each hospital, the preventive delegates (union representatives), the Chief Executive and the Human Resources Director.

COMMENT 4. If the process does not involve any real changes in work habits or work organization, it may be the reason why there was no significant changes in the results produced.

RESPONSE: Thank you for your comment. Although the majority of the proposed measures were implemented, it was not feasible to implement relevant other measures, especially the most expensive measures and measures that involved staff recruitment. Perhaps these implementation problems may explain the presented results. However, important changes in the work organization occurred due to the intervention. We have included this explanation in the Discussion section. 

[Page 18, paragraph 1, lines 372-377]

Preventive measures included structural, technical, organizational, training/information improvements in the workplace. Although the majority of proposed measures were implemented, It was not feasible to implement relevant other measures, especially the most expensive measures (e.g. staff recruitment, changing the rooms structure, etc.) and those that involved an expansion of the workforce. Perhaps these limitations explain the smaller than expected impact. 

COMMENT 5. In reporting the musculoskeletal pain, the "prevalence" rates were compared in terms of %. Is this just a "yes/no" answer in the questionnaire? Did you compare the pain rating for different body areas?

RESPONSE: For musculoskeletal pain, three answer categories were available “yes pain, yes discomfort, or no pain”. For the analysis, we dichotomized this variable (yes pain/discomfort, or no pain). Moreover, we estimated the pain rating before and after the intervention in the control group compared to the intervention group in seven anatomical sites: 1) neck, shoulders and upper back, 2) low back, 3) elbows, 4) hands, 5) legs, 6) knees and 7) feet. We have clarified this information in the Methods section, as follows:

[Page 9, paragraph 4, lines 192-198]

Information on pain or discomfort was collected in seven anatomical sites, i.e. (1) neck, shoulders and upper back, (2) low back, (3) elbows, (4) hands, (5) legs, (6) knees and (7) feet. We used the question “do you have pain or discomfort in this anatomical site?” with three possible answers (yes pain, yes discomfort, no pain). For the analysis of each anatomical site, the variable was dichotomized (yes pain or discomfort, no pain) to estimate the improvement of MSP over time, comparing the intervention group with the control group.

REVIEWER 2: A two arm randomized cluster controlled trial was conducted to evaluate an intervention to prevent or manage musculoskeletal pain in nurses. Compared to the control group, a decreased risk in shoulder, neck and back pain was observed in the intervention group. The manuscript is clearly written.

COMMENT 1: Abstract, Capitalize Cox.

RESPONSE: We have capitalized Cox in the abstract.

[Page 2, paragraph 2, lines 33-35]

The incidence of sickness absence was calculated through logistic regression and Cox proportional hazard modeling was used to analyze the effect of the intervention on sickness absence duration.

COMMENT 2: Line 193, Indicate the statistical testing method which achieves an 80% power and state if the alpha level is one- or two-sided.

RESPONSE: Thanks for the recommendation. The sample size was calculated through a likelihood-ratio test with a two-sided alpha level using the program Stata 13. We have incorporated this information in the sample size section: 

[Page 11, paragraph 1, lines 222-225] 

The sample size calculation was based on the primary outcome MSP, with an estimated prevalence of 80% for healthcare workers [3]. The target of the present study was to reduce the prevalence of MSP by 20% [47, 48]. We assumed alpha values (type I error) = 0.05 (two-sided), statistical power = 0.80 and an intraclass correlation coefficient (ICC) = 0.05. Using a likelihood-ratio test with these criteria, 82 workers were needed in the intervention and control group, respectively.

COMMENT 3: Table 1, State the statistical testing methods used to estimate the p-values.

RESPONSE: The statistical testing methods used to estimate the p-values were Chi Square Test or Fisher’s exact test when 20% or more cells of the contingency tables had an expected value lower than 5 or one value lower than 1. 

We have added the statistical testing method in Table 1, and we have specified this information in the statistical analysis section, as follows:

[Page 12, paragraph 2, lines 247-249] 

Baseline descriptive analyses were performed to compare the characteristics of the intervention and control groups using Chi Square Test, or Fisher’s exact test when 20% or more cells of the contingency tables had an expected value lower than 5 or one value lower than 1.

---

## [Editor Report · Decision Letter 1]

31 Oct 2019

Multifaceted intervention for the prevention and management of musculoskeletal pain in nursing staff: results of a cluster randomized controlled trial

PONE-D-19-15120R1

Dear Dr. Serra,

We are pleased to inform you that your manuscript has been judged scientifically suitable for publication and will be formally accepted for publication once it complies with all outstanding technical requirements.

With kind regards,

Andrea Martinuzzi

Academic Editor

PLOS ONE
---

## [Editor Report · Acceptance letter]

6 Nov 2019

PONE-D-19-15120R1 

Multifaceted intervention for the prevention and management of musculoskeletal pain in nursing staff: results of a cluster randomized controlled trial 

Dear Dr. Serra:

I am pleased to inform you that your manuscript has been deemed suitable for publication in PLOS ONE. Congratulations! Your manuscript is now with our production department. 

With kind regards,

on behalf of

Dr. Andrea Martinuzzi 

Academic Editor

PLOS ONE